# A Comprehensive Analysis of Transcriptomics and Proteomics Elucidates the Cold-Adaptive Ovarian Development of *Eriocheir sinensis* Farmed in High-Altitude Karst Landform

**DOI:** 10.3390/genes16091048

**Published:** 2025-09-06

**Authors:** Qing Li, Yizhong Zhang, Lijuan Li

**Affiliations:** 1College of Ecological Engineering, Guizhou University of Engineering Science, College Road, Bijie 551700, China; liqingdream@163.com (Q.L.);; 2Guizhou Key Laboratory of Plateau Wetland Conservation and Restoration, Bijie 551700, China

**Keywords:** *Eriocheir sinensis*, living temperature, ovarian development and maturation, transcriptomic, proteomic, FOXO3

## Abstract

**Background:** In high-altitude regions, sporadic two-year-old immature Chinese mitten crabs (*Eriocheir sinensis*) would overwinter and mature in their third year, developing into three-year-old crabs (THCs) with a cold-adaptive strategy. Compared to two-year-old crabs (TWCs) from low-altitude Jiangsu, THCs from Karst landform and high-altitude Guizhou exhibit significantly larger final size but lower gonadosomatic index (GSI) (*p* < 0.01). **Methods:** To elucidate the molecular mechanisms underlying this delayed ovarian development, integrated transcriptomic and proteomic analyses were conducted. **Results:** Results showed downregulation of PI3K-Akt and FoxO signaling pathways, as well as upregulation of protein digestion and absorption pathways. Differentially expressed proteins indicated alterations in mitochondrial energy transduction and nutrient assimilation. Integrated omics analysis revealed significant changes in nucleic acid metabolism, proteostasis, and stress response, indicating systemic reorganization in energy-nutrient coordination and developmental plasticity. **Conclusions:** The observed growth-reproductive inverse relationship reflects an adaptive life-history trade-off under chronic cold stress, whereby energy repartitioning prioritizes somatic growth over gonadal investment. Our transcriptomic and proteomic data further suggest a pivotal regulatory role for FOXO3 dephosphorylation in potentially coupling altered energy sensing to reproductive suppression. This inferred mechanism reveals a potential conserved pathway for environmental adaptation in crustaceans, warranting further functional validation.

## 1. Introduction

Water temperature serves as a master regulator factor for the physiology of poikilothermic animals and has a profound impact on the biochemical and physiological aspects of crustaceans. Beyond inducing disturbances in ionic and osmotic homeostasis [1], reducing oxygen diffusion and hemolymph circulation and limiting ATP production [2], cold stress also triggers energy reordering—shifting resources from reproduction to survival adaptation. Oogenesis is energy-intensive and requires the coordination of hormone signaling, vitellogenesis, and mitotic activity, all of which are thermosensitive processes [3]. The reproduction of crustaceans is controlled by the neuroendocrine systems through neuropeptides, hormones, and neurotransmitters [4], and temperature regulates their synthesis to enable them to lay eggs synchronously under optimal conditions. Studies have confirmed temperature-dependent gonadal maturation: when the temperature rises from 22 °C to 28 °C, the ovarian development of *Procambarus clarkii* accelerated [5]. *Procambarus llamasir* exhibited temperature-promoted egg laying between 15 °C and 25 °C [6]. In *Scylla serrate,* it was demonstrated that hatching eggs increased at higher temperatures [7]. Such thermal response peaks in reproductive diapause under cold stress, which is a complex phenotype characterized by developmental arrest, metabolic inhibition, and enhanced stress recovery ability [8]. This state is usually caused by environmental factors, including photoperiod, temperature, and nutrient availability [9]. Overall, these mechanisms establish cold-imposed diapause as an evolutionary strategy for energy conservation [10], yet the molecular drivers of ovarian energy redistribution in this process remain elusive.

FOXO3 is an evolutionarily conserved transcription factor and serves as a molecular link for energy distribution between reproduction and growth. In mammals, FOXO3 coordinates the activation of primordial follicles through dual mechanisms and simultaneously inhibits oxidative stress: SIRT1/SIRT3-mediated deacetylation, enhancing antioxidant capacity [11] and ROS-FOXO3-LC3/BNIP3 signaling regulates autophagy during ovarian aging [12]. Its phosphorylation-dependent nuclear cytoplasmic shuttle acts as a metabolic rheostat, controlling the transcriptional programs of energy homeostasis. It is crucial that FOXO3 dysregulation is the basis of mammalian ovarian insufficiency (POI) [13], which indicates the deep functional protection of reproductive plasticity in poikilothermic animals under abiotic stress. Despite this opposite effect, the mechanism by which cold stress reprograms gonadal energy distribution in crustaceans remains uncharacterized.

The Chinese mitten crab (*Eriocheir sinensis*) is a commercially vital aquaculture species in East Asia [14], prized for its high nutritional value and excellent flavor. Driven by market demand, in China, aquaculture has expanded from the middle-lower Yangtze River to inland regions, including high-altitude Karst areas of Guizhou Province, where superior overall availability of water resources supports artificial breeding. Due to the low water temperature in high-altitude regions, the farmed crabs are less prone to diseases, and their meat is of high quality, with a pleasant aroma and high umami intensity. Chinese mitten crabs are extensively farmed for commercial purposes and serve as a crucial source of income for local fishermen. Notably, crab ovarian development critically determines both commercial value (as the primary edible component) and reproductive fitness. Under typical rearing conditions, crabs reach sexual maturity after two years of reproductive development from megalopa larvae. However, in chronically cold environments characteristic of high-altitude (e.g., Guizhou, Qinghai) and high-latitude (e.g., Heilongjiang) regions, sporadic individuals delay maturation until the third year, forming distinct three-year-old crabs (THCs) [15,16,17,18]. Compared to two-year-old crabs from low-altitude regions (e.g., Jiangsu Province, J-TWC), three-year-old crabs from high-altitude Karst landform regions (e.g., Guizhou Province, G-THC) exhibit substantially larger final size but a reduced gonadosomatic index (GSI).

This prolonged life-history under cold-temperature stress indicates that temperature-mediated energy allocation trade-offs are beneficial to somatic cell growth rather than gonadal investment [1,19]. This trade-off is in line with the broader principle of temperature-dependent oxygen limitation, which governs physiological responses from the molecular to the organic level and can influence the energy efficiency optimization of poikilothermic animals, including crustaceans and zooplankton [2]. Although this framework explains the changes in species distribution and performance limitations caused by climate, the comprehensive molecular and physiological data elucidating the underlying mechanisms of thermal limitations, particularly concerning gonad development trade-offs in commercially important crustaceans under cold exposure in the wild, remain scarce.

Here, we integrated transcriptomic, proteomic and phenotypic analysis to compare three-year-old crabs from Guizhou (high-altitude, cold-adapted, G-THC) with two-year-old crabs from Jiangsu (low-altitude, warm-adapted, J-TWC). Our objective is to quantify cold-adapted ovarian investment changes through gonadosomatic index (GSI), exploring the regulatory network centered on FOXO3 through the integration of transcriptomics and proteomics, and to clarify the energy allocation mechanism that controls the growth-reproduction trade-off. This study will establish the first mechanistic framework for the cold adaptation life history strategy of decapod crustaceans, which is of great significance for selective breeding in sustainable aquaculture.

## 2. Materials and Methods

### 2.1. Sample Collection and Preparation

From February to March 2020, we purchased Megalopa larvae of *E. sinensis* from Jiangsu Suxie Aquaculture Co., Ltd. (Nantong, China), and transported them to Qingshan Green Water Ecological Development Co., Ltd., located in Bijie, Guizhou Province, China (27.15° N, 104.94° E; mean annual temperature 13 °C, elevation 1600 m ASL). The rearing at each stage of development is carried out in the breeding ponds. From December 2021 to March 2022, crabs slightly smaller than sexually mature adult crabs but with underdeveloped gonads were selected from the ponds and placed in oxygenated containers at 13 °C. They were transported to the laboratory within 24 h. Subsequently, under controlled aquaculture conditions, two-year-old immature crabs are continued to be raised to enter the sexual maturity stage, reaching the three-year-old stage around October 2022. Meanwhile, in October 2022, two-year-old mature female crabs were purchased from Yangcheng Lake, Suzhou, Jiangsu Province (31.38° N, 120.98° E; mean annual temperature 16 °C, elevation 2 m ASL). All specimens (15 female crabs in each group) were processed in October 2022. After fasting for 24 h, a comprehensive biometric assessment was conducted on the crabs. They were wiped dry with tissues, and the total wet mass was measured (±0.01 g). Crabs were lightly anesthetized in an ice bath for 5 min before dissection. After dissection of the frozen tray, ovarian tissues were quickly collected, and the wet mass of the ovaries was recorded (±0.01). Gonadosomatic index (GSI) calculation: GSI = (Ovarian mass/Total body mass) × 100. Meanwhile, the collected ovarian tissues (from five crabs) were placed in RNase-free sample tubes, snap-frozen in liquid nitrogen and stored at −80 °C for subsequent omics analyses. To ensure ethical compliance, this study received approval from the Animal Ethics Committee of the Guizhou University of Engineering Science (Approval No.: GUES2020-0001, date: 1 March 2019). All animal experiments were conducted in strict accordance with the guidelines for animal welfare.

### 2.2. RNA Isolation and Sequencing

In October 2022, using Trizol reagent and the modified method [20], total RNA was isolated from the ovarian tissues of five biologically independent female crabs in each experimental group (G-THC and J-TWC). RNA integrity was quantified using the RNA Nano 6000 Assay Kit on the Agilent 2100 Bioanalyzer System (Agilent Technologies, Santa Clara, CA, USA). Only samples with RNA integrity number (RIN) ≥ 8.0 and the rRNA ratio of 28S/18S > 1.8 were used for library construction. In the preparation of the library, oligonucleotide (dT)-conjugated magnetic beads were used to enrich polyadenylated mRNA. The purified mRNA was lysed at 94 °C for 5 min in First Strand Synthesis Reaction Buffer containing divalent cations. First-strand cDNA was synthesized with random hexamer primers and M-MuLV Reverse Transcriptase (RNaseH). Second-strand synthesis was then performed using DNA Polymerase I and dNTPs. The obtained double-stranded cDNA fragment was treated with exonuclease and polymerase in combination for blunt-end repair and then adenosinized at the 3′ end. The Illumina-compatible adapter with hairpin loop structure was ligated to the adenylated fragments using T4 DNA Ligase. cDNA fragments within the sizes ranging from 370 to 420 bp were selected using AMPure XP beads (Beckman Coulter, Beverly, MA, USA). The enriched fragments were amplified by PCR using Phusion High-Fidelity DNA polymerase, universal PCR primers, and index sequences. All library constructions and subsequent paired-end (2 × 150 bp) RNA sequencing were performed on the Illumina NovaSeq 6000 platform (Novogene Co., Beijing, China).

### 2.3. Transcriptome Assembly, Functional Annotation, and Data Analysis

Rigorous quality control of raw paired-end sequencing reads was carried out using FastQC. Adapter trimming, removal of reads containing ambiguous bases, and quality filtering were performed using Trimmomatic (v.0.39). High-quality clean reads were retained for subsequent analysis. De novo transcriptome assembly was performed using Trinity (v.2.13.2) with default parameters [21]. Assembled transcripts were functionally annotated through homology searches against multiple databases: Protein Family (Pfam), EuKaryotic Orthologous Groups (KOG), Swiss-Prot protein sequence database, Kyoto Encyclopaedia of Genes and Genomes (KEGG) and its associated Ortholog (KO) database, as well as Gene Ontology (GO) database. Gene expression levels were quantified as read counts mapped to the assembled transcriptome. The differential gene expression between G-THC and J-TWC was analyzed in R using edgeR package (v.3.22.5). The abundance of transcripts was normalized using the trimmed mean of M-values (TMM) method. The key DEG results were independently validated using DESeq2 (v.1.20.0) with default parameters (a false discovery rate (FDR)-adjusted *p*-value < 0.05). The genes identified as DEGs by edgeR and DESeq2 were retained for downstream analysis. The clusterProfiler R package (R studio version 4.5.1.) was used to identify the significantly enriched GO terms and KEGG pathways in DEGs [22].

### 2.4. Total Protein Extraction, Quantification, and Proteolysis

Ovarian tissues (~50 mg wet weight per sample) were homogenized on ice in 500 μL SDT lysis buffer (4% SDS, 100 mM Tris-HCl, 100 mM NaCl, pH = 8.0), and 1% (*v*/*v*) dithiothreitol (DTT; Sigma-Aldrich, St. Louis, MO, USA) was added. The homogenate was sonicated in an ice-water bath and centrifuged to clarify (12,000× *g*, 15 min, 4 °C). The supernatant was alkylated with 50 mM of iodoacetamide (IAM; Sigma-Aldrich) in the dark at room temperature for 1 h. Proteins were then precipitated by adding four volumes of ice-cold acetone and incubating at −20 °C for 2 h. Precipitates were pelleted, washed twice with 1 mL ice-cold acetone, air-dried for 5 min, and resolubilized in 8 M urea/100 mM triethylammonium bicarbonate (TEAB, pH = 8.5). Protein concentration was determined using the Bradford Protein Assay Kit (Beyotime Institute of Biotechnology, Haimen, China) with bovine serum albumin (BSA) standards. Protein integrity was verified by 12% SDS-PAGE, followed by Commassie Brilliant Blue R-250 staining. The sample concentration was normalized to 1.0 mg/mL using buffer solution. For protein hydrolysis, normalized protein samples were reduced with 10 mM DTT and alkylated with 20 mM IAM (at room temperature, 30 min in the dark). Then, the protein was digested with sequencing-grade trypsin at an enzyme-to-substrate ratio of 1:50 (*w*/*w*) in two steps: first, it was digested at 37 °C for 4 h, followed by a second overnight digestion that was carried out after supplementing trypsin (final ratio 1:25 *w*/*w*) and 1 mM CaCl_2_ at 37 °C. The digestion solution was acidified with formic acid (FA) to pH < 3.0, centrifuged (12,000× *g*, 5 min, room temperature) to remove insoluble substances, and desalinated using C18 solid-phase extraction (SPE) cylinder. The eluted peptides were freeze-dried and stored at −80 °C for liquid chromatography-tandem mass spectrometry (LC-MS/MS) analysis.

### 2.5. DDA Spectrum Library Construction

Lyophilized peptide samples were reconstituted in mobile phase A, separated at high pH (2% acetonitrile, pH adjusted to 10.0 with ammonium hydroxide), and centrifuged (12,000× *g*, 10 min, room temperature). The sample was fractionated on a C18 column using a Rigol L3000 HPLC system. A linear gradient was established between mobile phase A and mobile phase B (98% acetonitrile, pH 10.0 adjusted with ammonium hydroxide). Fractions were collected across the elution profile. All collected fractions were lyophilized and subsequently reconstituted in 0.1% (*v*/*v*) FA for LC-MS/MS analysis. Shotgun proteomic analysis was performed using two complementary LC-MS/MS platforms: EASY-nLC^TM^ 1200 UHPLC system (Thermo Fisher, Dreieich, Germany) online connected to Q Exactive^TM^ HF-X mass spectrometer, and the nanoElute UHPLC system (Bruker Daltonics, Bremen, Germany) coupled online to the timsTOF Pro2 mass spectrometer. For each platform, the LC-MS/MS workflow is as follows: reorganize the fraction supernatant, spiked with iRT retention time standard, and load it into the pre-column (homemade C18 Nano-Trap column). Peptides are separated on a C18 column using a linear gradient from 5% to 35% mobile phase B (typically over 90–120 min) in mobile phase A (0.1% FA in water) and mobile phase B (0.1% FA in acetonitrile). Survey scans were acquired over the range *m*/*z* 350 to 1500. The top 20 most intense precursor ions per cycle were isolated and fragmented. The raw data-dependent acquisition (DDA) data acquired from both platforms are processed collectively using specialized software to generate a comprehensive DDA spectrum library.

### 2.6. Data-Independent Acquisition (DIA) Mass Spectrometry Analysis

Chromatographic separation for DIA analysis was performed using a nanoElute UHPLC system (Bruker Daltonics, Germany) equipped with a Captive Spray ion source (Bruker Daltonics). The system operated with a spray voltage of 1.5 kV and maintained a full scan range from *m*/*z* 100 to 1700. Mass spectrometry acquisition was performed on the timesTOF Pro 2 (Bruker Daltonics). The MS1 resolution was set to 60,000 at *m*/*z* 200, with a full scan AGC target of 5 × 10^5^ and a maximum ion injection time of 20 ms. For MS2 fragmentation, peptides were analyzed at a resolution of 30,000 (at 200 *m*/*z)* using a normalized collision energy of 27% and AGC target of 1 × 10^6^.

### 2.7. Protein Identification, Quantification, and DEP Analysis

Raw MS/MS data was processed using the Spectronaut Pulsar platform. Database searching was performed based on the species-specific UniProt proteome database. Initial peptide spectrum matches (PSMs) were filtered based on the following criteria: PSM confidence score ≥ 99%, at least one unique peptide is required for protein identification, and the false discovery rate (FDR) ≤ 1%. Retention time alignment was calibrated using indexed retention time (iRT) standards. A Q-value cutoff of 0.01 was applied for precursor ion selection. Proteins exhibiting a fold change (FC) > 2.0 with statistical significance (*p* < 0.05) were defined as differentially expressed proteins (DEPs). The functional annotations of DEPs include Gene Ontology (GO) terms and InterPro domains (IPR), and they are conducted using InterProScan for integrated databases (Pfam, PRINTS, ProDom, SMART, ProSite, and PANTHER) [23]. Protein domain classification and pathway enrichment analysis were conducted using the Orthologous Groups (COG) and Kyoto Encyclopedia of Genes and Genomes (KEGG) databases, respectively. DEPs were visualized using volcano plots. Enrichment analysis for GO terms, IPR domains, and KEGG pathways was subsequently performed using the clusterProfiler package in R [24].

### 2.8. Integrated Analysis of Transcriptomics and Proteomics

To determine the molecular biological mechanisms underlying the observed phenotypic differences, we integrated the differentially expressed genes (DEGs) and differentially expressed proteins (DEPs) in the comparison between G-THC and J-TWC. clusterProfiler was used to conduct GO and KEGG pathway enrichment analyses on the DEGs and DEPs. The pathways with an adjusted *p*-value (FDR) <0.05 in both analyses were defined as significant co-enriched pathways. The co-enrichment pathways were functionally annotated based on enrichment significance −log_10_(FDR) and overlap ratio. To focus on the core DEPs, DEPs were filtered using strict criteria: |log_2_FC| > 1 and adjusted *p*-value (FDR) < 0.01. Protein–protein interaction (PPI) networks were predicted using the STRING database (v.10.0). The network is exported in default tab-delimited format for Cytoscape. PPI networks were imported, visualized, and analyzed using Cytoscape (v.3.9.1). Key centrality metrics (degree, betweenness centrality) were calculated using Cytoscape built-in tools. The key PPI modules were visualized in the comprehensive multi-panel graph.

### 2.9. Statistical Analysis

The Shapiro–Wilk test examines the normality of each set of data using SPSS 22.0 software, and GSI between the G-THC and J-TWC was compared using Independent samples *t*-test. Differences were considered significant at *p* < 0.05. All data were analyzed and plotted using the R studio version 4.5.1.

## 3. Results

### 3.1. Ovarian Development Comparison

Comparative assessment of reproductive phenotypes revealed significant divergence between G-THC and J-TWC groups (Figure 1). The GSI was markedly reduced in G-THC compared to J-TWC (*p* < 0.01), demonstrating a pronounced trade-off favoring somatic investment over gonadal allocation.

### 3.2. Comparative Transcriptomic Analysis of G-THC and J-TWC

Transcriptome analysis identified 2323 DEGs between G-THC and J-TWC groups (sampling in October 2022). The expression profile shows predominant downregulation, with 1429 genes downregulated and 894 genes upregulated. GO enrichment is significantly associated with transmembrane transporter activity (molecular function) and carbohydrate metabolic process (biological process) (Figure 2A). Downregulated DEGs were enriched in extracellular region (cellular component), carbohydrate metabolism, and transmembrane transporter activity (Figure 2B), whereas upregulated DEGs exhibited enrichment in transmembrane transporter activity, structural molecule activity, and cytoskeleton-dependent intracellular transport (Figure 2C). Key metabolic processes displayed asymmetric regulation: carbohydrate metabolism involves 18 upregulated genes and 35 downregulated genes, while transmembrane transporter activity shows 55 upregulated genes and 79 downregulated genes (Figure 2D).

KEGG pathway analysis identified 282 significantly enriched pathways in G-THC, including 245 downregulated pathways and 170 upregulated pathways (Note: the pathways may simultaneously contain up/down DEGs). Metabolic pathways involved the most DEGs (46 downregulated, 22 upregulated DEGs). It is worth noting that the PI3K-Akt signaling pathway (15 DEGs) and the FoxO signaling pathway (11 DEGs) exhibit significantly downregulated signaling cascades (Figure 2E). On the contrary, the protein digestion and absorption pathways were significantly upregulated (7 DEGs) (Figure 2F).

### 3.3. Comparative Proteomics Analysis of G-THC and J-TWC

After the spectral library was constructed in DDA, the DIA mass spectrometry was used for protein quantification. Rigorous quality control confirmed the reliability of the dataset. The coefficient of variation (CV) distribution between replicates exhibited low dispersion (median CV < 15%), and the cumulative frequency curve demonstrated rapid stability (Appendix A). Principle component analysis (PCA) showed significant separation between groups (G-THC vs. J-TWC), tight clustering within groups (Appendix A), and high experimental reproducibility. Comparative proteomics identified 771 DEPs between G-THC and J-TWC, among which 263 were upregulated and 508 were downregulated (Figure 3A, Appendix A). GO enrichment analysis (FDR < 0.05) showed that there were 303 significant terms in the three ontologies. In biological process, 136 terms are enriched in carbohydrate derivative metabolism, protein folding regulation, and nucleoside triphosphate metabolism. Molecular function annotation identified 102 terms mainly related to ion binding (particularly cation and metal ion). The cellular component mapping indicated that DEPs are primarily enriched in cytoplasm and cytoplasmic part (Figure 3B). Remarkably, up- or down-regulated DEP clustering showed that the number of upregulated and downregulated DEPs enriched in the binding term was the highest (Figure 3C).

KEGG analysis of the top 20 enriched pathways revealed differences in metabolic reprogramming (Figure 3D). Global DEP enrichment is associated with mitochondrial energy transduction (oxidative phosphorylation), nutrient assimilation (protein digestion and absorption), and growth (PI3K-Akt signaling pathway). The upregulated DEPs dominate the biosynthetic process, including lipid anabolism (fatty acid biosynthesis) and the regulation of cellular energy homeostasis (AMPK signaling pathway). Conversely, downregulated DEPs exhibit catabolic tendencies, particularly in terms of vesicular transport regulation (endocytosis), and angiogenesis-related signal transduction (VEGF signaling pathway) (Appendix A).

### 3.4. Integrated Analysis of Transcriptomic and Proteomic Profiles

Comprehensive transcriptomic–proteomics analysis identified 27 GO terms and 116 KEGG pathways with coordinated expression patterns (Appendix A). The core functional convergence of the top 15 shared GO terms (Figure 4A) spans five modules: nucleic acid dynamics (DNA metabolic process), protein homeostasis (protein folding and unfolded protein binding), cellular structure (extracellular region, extracellular matrix and intracellular), stress adaptation (response to stress and mitochondrion), and metabolic regulation (carbohydrate metabolic process and biosynthetic process). The integration at the pathway level revealed regulatory differences across the entire system (Figure 4B), mainly dominated by the energy-nutrient axis (FoxO signaling pathway, PI3K-Akt signaling pathway, and oxidative phosphorylation), developmental plasticity (Wnt signaling pathway and lysine degradation), pathogen interactions (epithelial cell signaling in *Helicobacter pylori* infection), and oncogenic characteristics (non-small cell lung cancer).

Through topology-based examination of five critical KEGG pathways (FoxO signaling pathway, protein digestion and absorption, PI3K-Akt signaling pathway, starch and sucrose metabolism, and metabolic pathways), we pinpointed 42 key interactors and 19 hub proteins (Figure 5). Upregulated biosynthetic hubs: GPI (glucose-6-phosphate isomerase, the gatekeeper of glycolysis), FOXO3 (stress-responsive transcription factor), B3GALT1/A4GALT (glycosphingolipid biosynthesis), and NIT2 (nitrogen metabolic regulator). Conversely, downregulated catabolic hubs: hydrolases (TREH: trehalase, MGAM: maltase, GAA: glucosidase), metabolic bypass (PCK: phosphoenolpyruvate carboxykinase), stress signals (SGK1/2: osmoregulatory kinases, ATM-PHLPP: DNA damage checkpoint), and turnover mediators (BNIP3: mitochondrial autophagy, ANPEP: aminopeptidase).

## 4. Discussion

The regulation of ovarian maturation in crustaceans is highly diverse and involves both endogenous and external environmental factors. Among these, water temperature serves as a key external factor. During breeding seasons, when the water temperature remains above 22 °C, female oriental prawns and crawfish enter a period of rapid development characterized by shortened ovarian maturation cycles [5,25]. In contrast, lower water temperature decelerates gonadal development, potentially induces diapause, delays sexual maturation, and extends lifespan [15,16,17]. In *Drosophila melanogaster*, low-temperature exposure can lead to reproductive quiescence, delayed senescence, extended lifespan, and enhanced stress resistance. The incidence of diapause in natural populations shows a distinct latitudinal cline across the eastern United States [26]. In *E. sinensis*, although individuals from high-altitude environments exhibit prolonged growth periods and larger body sizes, this advantage correlates with a significant reduction in gonadal investment, reflected by a decreased GSI and compromised fecundity, which may ultimately undermine population sustainability [27]. This inverse relationship between growth and reproduction reflects an evolutionary trade-off under cold stress, wherein energy redistribution is prioritized toward cold adaptation rather than gametogenesis [10]. A number of functional genes (including *CERK*, heat-stress related genes, *AMPK*, *FOXOs*, and gene-encoded metabolic-related enzymes) have been implicated in thermal adaptation [19,28,29].

### 4.1. Metabolic Reprogramming for Cold Adaptation

Glucose-6-phosphate isomerase (GPI) is a key glycolytic enzyme that catalyzes the interconversion between glucose-6-phosphate (G6P) and fructose-6-phosphate (F6P) [30], and GPI was significantly upregulated in the G-THC group under chronic cold stress. This metabolic recombination redirects carbon fluxes through two synergistic adaptations. Enhanced reverse catalysis (F6P-G6P) guided the substrate towards chitin biosynthesis [31], which a critical exoskeleton modification for thermal insulation in crustaceans [32]. Meanwhile, GPI-mediated suppression of trehalase (TREH) reduces trehalose hydrolysis, promotes intracellular trehalose accumulation, and alleviates cold-induced cellular dehydration [33]. Therefore, GPI exemplifies functional plasticity in ectotherm thermal adaptation, resolving the energy allocation conflict between basal metabolism and stress defense through the directional metabolic flux control from glycolysis to anabolism and the allocation of osmolyte resource. This dual-action model positions GPI as the strategic target for breeding *E. sinensis* strains with cold resistance without compromising stress tolerance.

### 4.2. FOXO3 Phosphorylation Coordinates the Trade-Off of Energy Between Growth and Reproduction Under Cold Stress

Under cold stress, cellular energy homeostasis depends on the antagonistic effect of the PI3K-Akt-AMPK signaling pathway module. Akt promotes anabolism (such as protein synthesis), while AMPK activates catabolism (such as autophagy) to compensate for energy deficiency [28,34,35]. Under long-term chronic cold exposure, in order to maintain AMPK activation, the energy for ovarian maturation is transferred by inhibiting the PI3K-Akt-mTOR signaling pathway module—a conserved trade-off, prioritizing stress adaptation over reproduction [36,37]. Comprehensive transcriptomic and proteomic analyses determined that FOXO3 is the central regulator coordinating this energy distribution. Specifically, Akt-mediated phosphorylation at Thr32, Ser253, and Ser315 sites triggers cytoplasmic isolation of FOXO3, inhibiting autophagy genes. Conversely, AMPK-mediated phosphorylation at Ser413, Ser439 and Ser585 sites promotes nuclear translocation of FOXO3 and activates stress-response genes (Figure 6).

In the cold-adapted G-THC ovaries, the molecular adaptation strategy involves three interrelated key components: ATM-PHLPP dysregulation, FOXO3 bilocalization status, and metabolic initiation. Firstly, ATM-PHLPP dysregulation inhibits ATM activity and weakens AMPK activation, while the instability of PHLPP impairs Akt dephosphorylation [38]. Secondly, this dysregulation establishes a dual FOXO3 state. The cytoplasmic retention of Akt-phosphorylated FOXO3 limits autophagic degradation (as evidenced by decreased GAA activity) [39], thereby protecting developing oocytes from catabolic damage. Meanwhile, the nuclear translocation of AMPK-phosphorylated FOXO3 inhibits the expression of vitellogenin, leading to follicular atresia. Thirdly, metabolic initiation is obvious: GPI-mediated glycolysis inhibition increases the AMP/ATP ratio, amplifies AMPK activation and subsequent FOXO3 phosphorylation and signaling transduction. Overall, this phosphorylation-dependent switch dynamically allocates cellular energy: nuclear FOXO3 redirects resources towards stress defense mechanisms, while cytoplasmic FOXO3 preserves the integrity of gametes. This complex regulatory network optimizes the fitness of organisms under low-temperature constraints [40,41].

## 5. Conclusions

This study combined the GSI measurement results with the transcriptomic and proteomic analyses of the Chinese mitten crab (*E. sinensis*) living in high-altitude cold (G-THC) and low-altitude warm (J-TWC) habitats. The results showed that the GSI of the G-THC ovary was significantly decreased (*p* < 0.01), indicating that reproductive investment was impaired under cold stress. The convergent enrichment of DEGs and DEPs centered on three core adaptive modules: the energy-nutrition sensing axis, the developmental plasticity pathway, and the pathogen interaction network. Overall, these findings speculate on a metabolic trade-off where cold adaptation is prioritized over reproduction, redirecting the energy resources for ovarian development towards stress defense mechanisms. The core of this energy recalibration is the spatiotemporal phosphorylation of FOXO3, which acts as a molecular switch controlling the distribution of carbohydrate fluxes between somatic cell growth and reproductive processes. However, further research on the upstream regulatory factors of FoxO3 dynamics is needed to fully clarify the molecular mechanism of this low-temperature adaptation regulation.

## Figures and Tables

**Figure 1 genes-16-01048-f001:**
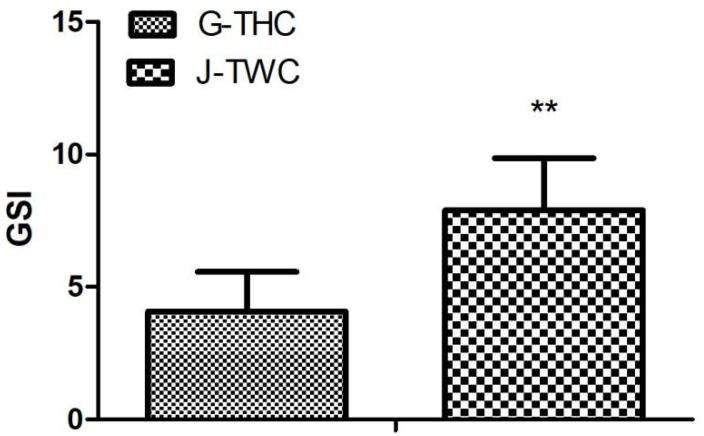
GSI between G-THC and J-TWC groups. GSI values presented as means ± SEM (n = 15). “**” *p* < 0.01 by *t*-tests.

**Figure 2 genes-16-01048-f002:**
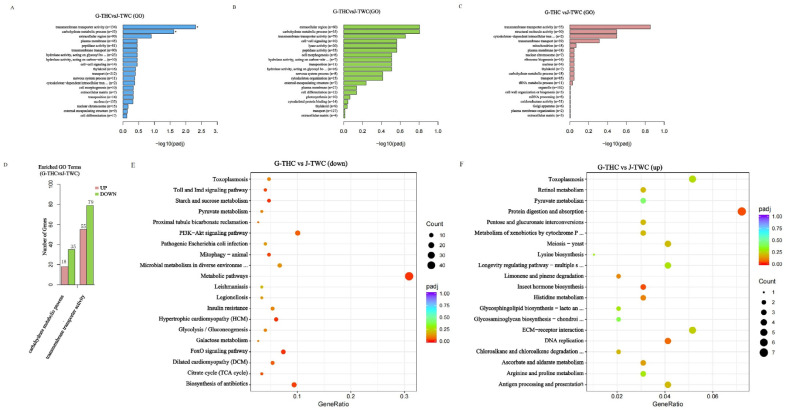
Transcriptomic profiling reveals distinct biological processes and pathways modulated in G-THC compared to J-TWC. (**A**) Gene Ontology (GO) enrichment analysis of all differentially expressed genes (DEGs), showing the top significantly enriched terms across biological process, cellular component, and molecular function categories. (**B**,**C**) GO enrichment analysis of downregulated (**B**) and upregulated (**C**) DEGs, highlighting the specific functional categories most affected by the suppression or induction of gene expression, respectively. (**D**) Counts of upregulated and downregulated DEGs within two key significantly enriched GO terms: carbohydrate metabolic process and transmembrane transporter activity. The opposing trends suggest a fundamental shift in energy metabolism and cellular transport in G-THC. (**E**,**F**) KEGG pathway enrichment analysis (top 20 pathways) for downregulated (**E**) and upregulated (**F**) DEGs. The downregulated genes are primarily involved in metabolic pathway, biosynthesis of antibiotics and PI3K-Akt signaling pathway, while upregulated genes are enriched in pathways related to protein digestion and absorption and DNA replication, indicating a potential trade-off between growth and defense response in the G-THC. “*” means significantly different (*p* < 0.05).

**Figure 3 genes-16-01048-f003:**
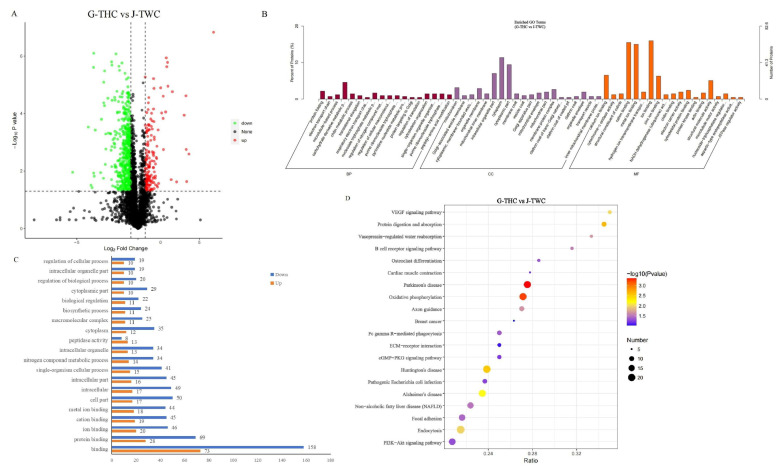
Proteomic analysis identifies key functional shifts in energy and stress response in G-THC. (**A**) Volcano plot of differentially expressed proteins (DEPs) between G-THC and J-TWC groups. Proteins with significant upregulation (red) and downregulation (green) are highlighted (|log2 FC| > 2, *p*-value < 0.05). (**B**) Gene Ontology (GO) functional classification of all DEPs, showing their distribution across biological process, cellular component, and molecular function categories. This overview indicates that metabolic and catalytic processes are predominantly affected. (**C**) The top 20 most significantly enriched GO terms for up-(orange) and down-regulated (blue) DEPs. Significantly downregulated and upregulated DEPs are both related to binding and protein binding terms. (**D**) The top 20 most significantly enriched KEGG pathways for DEPs. The enrichment results corroborate the GO analysis, highlighting severe suppression of oxidative phosphorylation, pathogenic Escherichia coli infection and endocytosis pathways, and activation of pathways related to protein digestion and absorption and fatty acid biosynthesis pathways, reinforcing the model of induced stress adaptation in the G-THC group.

**Figure 4 genes-16-01048-f004:**
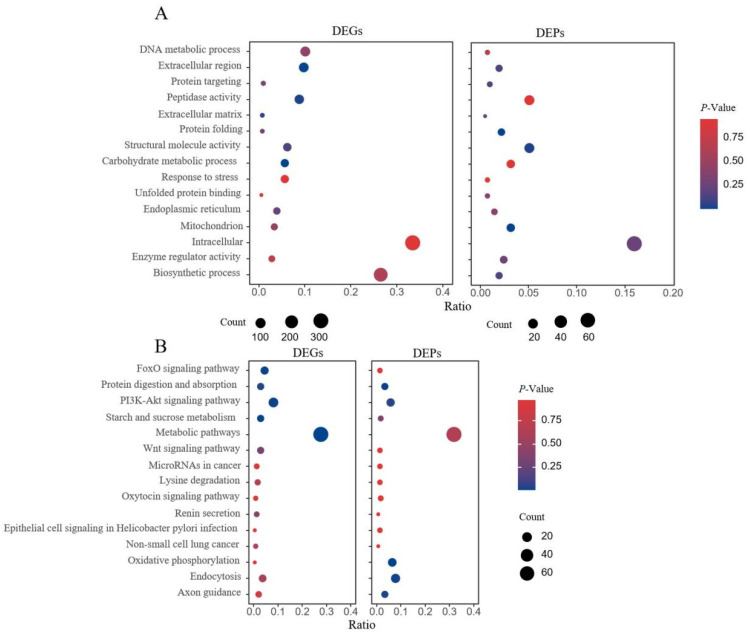
Integrated transcriptomic and proteomic analysis reveals coordinated regulation of key metabolic and stress-responsive pathways in G-THC. (**A**) Gene Ontology (GO) enrichment analysis (top 15 terms) for differentially expressed genes (DEGs) and proteins (DEPs) common to both omics. The significant concordance in terms related to response to stress, peptidase activity, carbohydrate metabolic process, and mitochondrion underscores a robust, multi-layered reprogramming of the central metabolism and defense systems. (**B**) KEGG pathway enrichment analysis (top 15 pathways) of concordant DEGs and DEPs. Key pathways such as FoxO signaling pathway, PI3K-Akt signaling pathway, oxidative phosphorylation, protein digestion and absorption, starch and sucrose metabolism, Wnt signaling pathway, and endocytosis are prominently enriched, highlighting a strategic shift in resource allocation from primary growth metabolism towards pathways essential for stress mitigation and cellular homeostasis.

**Figure 5 genes-16-01048-f005:**
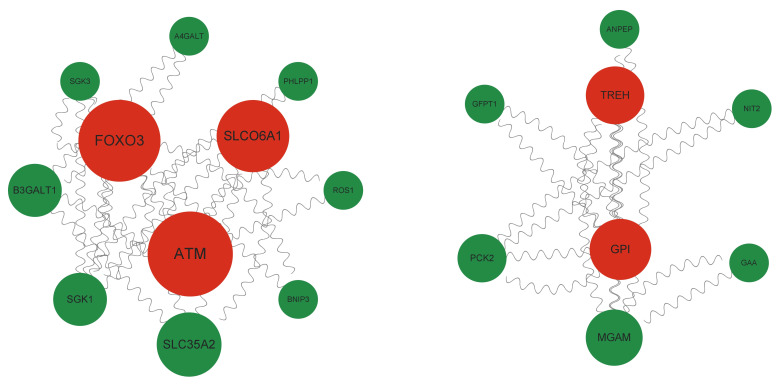
Protein–protein interaction (PPI) network analysis identifies hub proteins central to the stress adaptation response in G-THC. The PPI network was constructed using proteins encoded by differentially expressed genes (DEGs) that were also enriched in key GO terms and KEGG pathways from the integrated transcriptomic and proteomic analysis. Hub proteins were computationally identified based on their high betweenness centrality score, a measure of their importance as regulatory bottlenecks in the network. Proteins with high scores are depicted as red nodes, while those with low scores are in green. These hub proteins, which include stress-responsive transcription factor (FOXO3), DNA damage checkpoint (ATM), trehalase (TREH) and the gatekeeper of glycolysis (GPI), are predicted to function as critical molecular regulators orchestrating the observed shift from growth to defense metabolism.

**Figure 6 genes-16-01048-f006:**
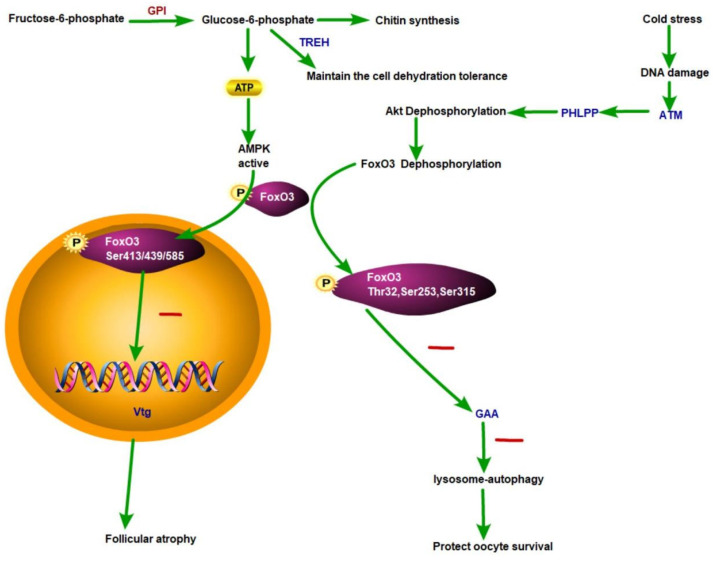
Schematic representation of FOXO3 activity regulation by its subcellular localization and regulators in allocation toward growth or reproduction under chronic cold adaptation.

## Data Availability

Raw RNA-Seq data for the G-THC and J-TWC groups are available at the NCBI SRA under accession number: PRJNA1280669. The mass spectrometry proteomics data have been deposited to the ProteomeXchange Consortium (https://proteomecentral.proteomexchange.org/cgi/GetDataset?ID=PXD065485, accessed on 7 August 2025) via the iProX partner repository (https://www.iprox.cn/page/project.html?id=IPX0012414000, accessed on 7 August 2025) with the dataset identifier PXD065485.

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
