# Peer review of "A Comprehensive Analysis of Transcriptomics and Proteomics Elucidates the Cold-Adaptive Ovarian Development of Eriocheir sinensis Farmed in High-Altitude Karst Landform"

_genes, 2025, doi:10.3390/genes16091048_

Round 1
Reviewer 1 Report
Comments and Suggestions for Authors
General Comment#
The manuscript entitled "A Comprehensive Analysis of Transcriptomics and Proteomics Elucidate the Cold-adaptive Ovarian Development in Eriocheir 3 sinensis of High Altitude Karst Landform" by Qing et. al presents an integrated transcriptomic and proteomic analysis to investigate cold-adaptive ovarian development in Eriocheir sinensis from high-altitude karst landforms. The authors convincingly demonstrate that three-year-old high-altitude crabs exhibit larger body size but a reduced gonadosomatic index compared to two-year-old lowland crabs, and link these phenotypic differences to the downregulation of PI3K–Akt and FoxO signaling, alongside metabolic reprogramming. The study is novel, well-designed, and has the potential to be impactful for both aquaculture and ecological physiology research.
This study offers valuable mechanistic insights into cold-adaptive reproductive trade-offs in E. sinensis. With more straightforward presentation, condensed methods, and improved focus in discussion, it will make a substantial contribution to aquaculture genomics and environmental adaptation research. I recommend the author's minor revision – see the comments under 'points for improvement".
Strengths
- Novelty & Significance
- First integrative omics study to dissect cold-adaptive ovarian development in a commercially important crustacean.
- Links physiological trade-offs (growth vs. reproduction) to molecular mechanisms (FOXO3 phosphorylation dynamics).
- Findings have practical relevance in aquaculture (breeding of cold-tolerant but reproductively fit strains).
- Comprehensive Methodology
- High-quality RNA-seq (NovaSeq 6000, RIN ≥ 8.0) and DIA-based proteomics ensure the generation of robust data.
- Rigorous QC (e.g., PCA clustering, FDR thresholds) supports data reliability.
- Integration of DEGs and DEPs into pathway analysis adds depth.
- Interpretation & Conceptual Advance
- Clear articulation of a life-history trade-off under chronic cold stress.
- Identification of FOXO3 as a central regulatory hub is biologically compelling.
Points for Improvement
FOXO3 is introduced late in the study; it should be highlighted earlier as the study's central hypothesis.
Ethical approval is noted, but could be expanded (permit numbers, aquaculture guidelines) if applicable.
Some figures under two could be illustrated as panels. Figures require clearer legends (biological context, not just methods/statistics).
Volcano plots and enrichment bar plots are informative, but a summary figure/table of key DEGs/DEPs with fold-changes would improve readability.
In the discussion section, the authors overemphasize FOXO3 relative to other findings (e.g., the potential for expanding GPI/trehalose metabolism could be explored). It requires a clearer comparative context: how do these findings relate to other cold-adapted crustaceans or model poikilotherms?
Several typographical errors and awkward phrases (e.g., "purchased (Line 105)," "factors (Line 78),") require polishing. Sentences are often long and complex; readability could be improved.
Author Response
Comments1
The manuscript entitled "A Comprehensive Analysis of Transcriptomics and Proteomics Elucidate the Cold-adaptive Ovarian Development in Eriocheir 3 sinensis of High Altitude Karst Landform" by Qing et. al presents an integrated transcriptomic and proteomic analysis to investigate cold-adaptive ovarian development in Eriocheir sinensis from high-altitude karst landforms. The authors convincingly demonstrate that three-year-old high-altitude crabs exhibit larger body size but a reduced gonadosomatic index compared to two-year-old lowland crabs, and link these phenotypic differences to the downregulation of PI3K–Akt and FoxO signaling, alongside metabolic reprogramming. The study is novel, well-designed, and has the potential to be impactful for both aquaculture and ecological physiology research.
This study offers valuable mechanistic insights into cold-adaptive reproductive trade-offs in E. sinensis. With more straightforward presentation, condensed methods, and improved focus in discussion, it will make a substantial contribution to aquaculture genomics and environmental adaptation research. I recommend the author's minor revision – see the comments under 'points for improvement".
Strengths
Novelty & Significance
First integrative omics study to dissect cold-adaptive ovarian development in a commercially important crustacean.
Links physiological trade-offs (growth vs. reproduction) to molecular mechanisms (FOXO3 phosphorylation dynamics).
Findings have practical relevance in aquaculture (breeding of cold-tolerant but reproductively fit strains).
Comprehensive Methodology
High-quality RNA-seq (NovaSeq 6000, RIN ≥ 8.0) and DIA-based proteomics ensure the generation of robust data.
Rigorous QC (e.g., PCA clustering, FDR thresholds) supports data reliability.
Integration of DEGs and DEPs into pathway analysis adds depth.
Interpretation & Conceptual Advance
Clear articulation of a life-history trade-off under chronic cold stress.
Identification of FOXO3 as a central regulatory hub is biologically compelling.
Points for Improvement
- FOXO3 is introduced late in the study; it should be highlighted earlier as the study's central hypothesis.
Response: Thanks. We have revised about FOXO3 part in the Introduction section according to suggestions.
- Ethical approval is noted, but could be expanded (permit numbers, aquaculture guidelines) if applicable.
Response: Thank you. We have added the Approval No. in the “Sample collection and preparation” according to the suggestions.
- Some figures under two could be illustrated as panels. Figures require clearer legends (biological context, not just methods/statistics).
Response:Thanks for your suggestions. We have revised the figures and legends in the whole manuscript.
- Volcano plots and enrichment bar plots are informative, but a summary figure/table of key DEGs/DEPs with fold-changes would improve readability.
Response:Thanks for your suggestion. We have added the table of DEPs with fold-changes as supplementary table1.
- In the discussion section, the authors overemphasize FOXO3 relative to other findings (e.g., the potential for expanding GPI/trehalose metabolism could be explored). It requires a clearer comparative context: how do these findings relate to other cold-adapted crustaceans or model poikilotherms?
Response: Thanks.We have added the related background in the discussion according to suggestions.
- Several typographical errors and awkward phrases (e.g., "purchased (Line 105)," "factors (Line 78),") require polishing. Sentences are often long and complex; readability could be improved.
Response:Thanks for your comments. We have revised the typographical errors and sentences to readability in the whole manuscript.

Reviewer 2 Report
Comments and Suggestions for Authors
This manuscript focused on the effect of living temperature on the crab ovary development. I found the study well presented, justified and discussed. Notably, a wide range of complementary analytical tools were employed, classical and very advanced. I think it can be accepted for publication provided some minor aspects are clarified. I would mention the following:
Title:
I think the word “farmed” could be included.
Abstract
It could be shortened to agree with the journal requirements (ca. 200 words).
Indicate more concretely what was compared in the study.
Keywords
Replace low temperature with living temperature. Include maturation.
Material and methods
A wide range of complementary analyses were carried out. Especially valuable are those analyses related to transcriptomics and proteomics.
No statistical analysis is mentioned. I think this is the main aspect that ought to be performed. Were there replicates carried out ? Provide the number of individuals in the different pools employed.
Results
As a food is concerned and because of the great economic significance of the present crab species, some comment and discussion regarding the effect of temperature living condition on commercial availability and sensory acceptance by consumers could be included.
Can the effect of living temperature have a similar behaviour on the ovarian development of other marine invertebrate species ? Does previous research include some related conclusions ? Please, include some comments on this if available in bibliography. If not available, including such comment may provide some additional value to the manuscript.
Author Response
Comments 2
This manuscript focused on the effect of living temperature on the crab ovary development. I found the study well presented, justified and discussed. Notably, a wide range of complementary analytical tools were employed, classical and very advanced. I think it can be accepted for publication provided some minor aspects are clarified. I would mention the following:
Title:
- I think the word “farmed” could be included.
Response: Thanks for your suggestions. We have revised the Title.
- Abstract
It could be shortened to agree with the journal requirements (ca. 200 words).
Indicate more concretely what was compared in the study.
Response:Thanks. We have revised the Abstract according to the suggestions.
- Keywords
Replace low temperature with living temperature. Include maturation.
Response: Thanks for your comments. We have revised according to suggestions.
- Material and methods
A wide range of complementary analyses were carried out. Especially valuable are those analyses related to transcriptomics and proteomics.
No statistical analysis is mentioned. I think this is the main aspect that ought to be performed. Were there replicates carried out ? Provide the number of individuals in the different pools employed.
Response: Thanks for comments. We have added the statistical analysis and the number of individuals of different groups in the Material and methods section.
- Results
As a food is concerned and because of the great economic significance of the present crab species, some comment and discussion regarding the effect of temperature living condition on commercial availability and sensory acceptance by consumers could be included.
Response:Thanks. We have added related part in Introduction section according to suggestions.
- Can the effect of living temperature have a similar behaviour on the ovarian development of other marine invertebrate species ? Does previous research include some related conclusions ? Please, include some comments on this if available in bibliography. If not available, including such comment may provide some additional value to the manuscript.
Response:Thanks for suggestion. We have added related conclusions in the key bibliography.
